# Integrative Analysis of Metabolome and Transcriptome of Carotenoid Biosynthesis Reveals the Mechanism of Fruit Color Change in Tomato (*Solanum lycopersicum*)

**DOI:** 10.3390/ijms25126493

**Published:** 2024-06-12

**Authors:** Jiahui Hu, Juan Wang, Tayeb Muhammad, Tao Yang, Ning Li, Haitao Yang, Qinghui Yu, Baike Wang

**Affiliations:** 1College of Horticulture, Xinjiang Agricultural University, Urumqi 830052, China; 2Key Laboratory of Genome Research and Genetic Improvement of Xinjiang Characteristic Fruits and Vegetables, Institute of Horticulture Crops, Xinjiang Academy of Agricultural Sciences, Urumqi 830000, China; drjuanwang@126.com (J.W.);

**Keywords:** tomato, fruit ripening, metabolome, transcriptome, carotenoids, lycopene, ethenyl

## Abstract

Tomato fruit ripening is accompanied by carotenoid accumulation and color changes. To elucidate the regulatory mechanisms underlying carotenoid synthesis during fruit ripening, a combined transcriptomic and metabolomic analysis was conducted on red-fruited tomato (WP190) and orange-fruited tomato (ZH108). A total of twenty-nine (29) different carotenoid compounds were identified in tomato fruits at six different stages. The abundance of the majority of the carotenoids was enhanced significantly with fruit ripening, with higher levels of lycopene; (E/Z)-lycopene; and α-, β- and γ-carotenoids detected in the fruits of WP190 at 50 and 60 days post anthesis (DPA). Transcriptome analysis revealed that the fruits of two varieties exhibited the highest number of differentially expressed genes (DEGs) at 50 DPA, and a module of co-expressed genes related to the fruit carotenoid content was established by WGCNA. qRT-PCR analysis validated the transcriptome result with a significantly elevated transcript level of lycopene biosynthesis genes (including *SlPSY2*, *SlZCIS*, *SlPDS*, *SlZDS* and *SlCRTSO2*) observed in WP190 at 50 DPA in comparison to ZH108. In addition, during the ripening process, the expression of ethylene biosynthesis (*SlACSs* and *SlACOs*) and signaling (*SlEIN3* and *SlERF1*) genes was also increased, and these mechanisms may regulate carotenoid accumulation and fruit ripening in tomato. Differential expression of several key genes in the fruit of two tomato varieties at different stages regulates the accumulation of carotenoids and leads to differences in color between the two varieties of tomato. The results of this study provide a comprehensive understanding of carotenoid accumulation and ethylene biosynthesis and signal transduction pathway regulatory mechanisms during tomato fruit development.

## 1. Introduction

Tomato (*Solanum lycopersicum*) is one of the most important horticultural crops in the world and an essential fruit in the human diet [1]. Tomato fruits are rich in minerals, vitamins (ascorbic acid and vitamin A), phenolic compounds (phenolic acids and flavonoids) and carotenes (lycopene, β-carotene and phytoene), which accumulate as fruits ripen [2]. Fruit ripening is a complex biological process, and as a primary model of climacteric fruit, the color, texture, flavor and aroma of the fruit flesh undergo significant changes during the ripening period [3]. In tomato, the change from green to red coloration is the most visual indicator of fruit ripening, and this shift is directly related to carotenoid biosynthesis [4]. Carotenoids are essential nutrients in tomato fruits, and their content and composition influence not only the color, flavor and marketability of the tomato fruit but also its potential health benefits for humans, such as anticancer, blood pressure reduction, immunity enhancement and anti-aging, which are the key quality characteristics of the crop [5]. Geranylgeranyl pyrophosphate (GGPP) is a direct precursor of carotenoid biosynthesis, and the condensation of two molecules of GGPP forms carotene 15-cis-phytoene through the phytoene synthase enzyme (PSY) catalytic activity [6]. PSY is the central enzyme that regulates the total amount of carotenoids accumulated in plant tissues, and changes in PSY activity significantly affect the metabolism and downstream compounds of the carotenoid pathway [7]. Subsequently, four enzymes, phytoene desaturase (PDS), ζ-carotene desaturase (ZDS), prolycopene isomerase (CRTISO) and ζ-carotene isomerase (ZISO), desaturate 15-cis-phytoene and convert it to trans-lycopene [8,9]. The synthesized trans-lycopene is transformed into δ-carotene or γ-carotene by lycopene δ-cyclase (LCYE) or lycopene β-cyclase (LCYB), respectively [10]. LCYB also catalyzes the conversion of δ-carotene to α-carotene and γ-carotene to β-carotene. α-carotene is ultimately converted to lutein through hydroxylation, which is catalyzed by ε-cyclic carotene hydroxylase (CHYE) and β-cyclic carotene hydroxylase (CHYB) [11]. Similarly, the four oxidation reactions regulated by CHYB, zeaxanthin epoxidase (ZEP), violaxanthin de-epoxidase (VDE) and neoxanthin synthase (NXS) lead to the conversion of β-carotene to zeaxanthin, antheraxanthin, violaxanthin and neoxanthin [12]. Finally, violaxanthin and neoxanthin are cleaved by 9-cis epoxycarotenoid dioxygenase (NCED), which is a key enzyme in ABA biosynthesis [13].

Tomato is a typical climacteric fruit; it undergoes high respiration and ethylene production during fruit development and ripening [14]. Ethylene is a phytohormone that positively regulates fruit ripening and accelerates the accumulation of lycopene content [14,15]. In tomato fruit, ethylene biosynthesis is catalyzed by two main enzymes: firstly, SAM (S-adenosylmethionine) is catalyzed by ACS (1-aminocyclopropane- 1-carboxylic acid synthase) to produce ACC (1-aminocyclopropane-1-carboxylic acid), and then ACC is converted to ethylene by ACO (ACC oxidase) activity [16]. Ethylene-induced fruit ripening is associated with a transcriptional cascade of ethylene-responsive genes, including ethylene-insensitive (EIN) and ethylene-responsive factor (ERF) genes [17,18]. EIN is a small family of transcription factors that could bind directly to the conserved EIN3 binding sequence (EBS) [19,20]. In contrast to EIN, ERF is a larger family of plant transcription factors, and it plays a crucial and wide-ranging role in plant development, particularly in fruit ripening, hormone signaling and stress responses [21,22,23]. However, the specific molecular regulatory mechanisms of ethylene-responsive genes during fruit ripening require further verification and analysis.

In the present study, we collected and compared two tomato varieties with substantial differences in fruit color at the ripening stage. The fruit color of WP190 is red, and WP190 is an introduced variety from Israel containing a high lycopene content compared to ZH108, a native orange tomato. To elucidate the underlying mechanisms for the observed differences in carotenoid metabolism and fruit color between the two tomato varieties during fruit ripening, transcriptome sequencing and metabolomic analysis were conducted at six different developmental and ripening stages. The combined data from the metabolome and transcriptome were subjected to WGCNA analysis, which allowed for the identification of co-expressed genes and transcription factors related to carotenoid metabolism at different fruit developmental stages between the two varieties. Moreover, the expression trends of genes involved in the carotenoid synthesis pathway, including *SlPSY2*, *SlPDS*, *SlZCIS*, *SlZDS*, *SlCRTSO2* and *SlLCYB2*, were analyzed at different stages in the two tomato varieties, and the expression profiles of genes involved in ethylene biosynthesis and signaling were examined at the same time, including *SlACS1*, *SlACS4*, *SlACO2*, *SlACO4*, *SlEIN3* and *SlERF1*. The findings of the current study reveal that the key genes involved in carotenoid and ethylene synthesis play a pivotal role in regulating the ripening and coloration of tomato fruits.

## 2. Results

### 2.1. Metabolic Analysis of Carotenoid Synthesis in Tomato Fruits

We collected fruits of two cultivated tomato varieties, WP190 (introduced from Israel) with high red pigmentation and ZH108 (introduced from Xinjiang, China), at different developmental and ripening stages. The phenotypes of eight fruit stages from 5 to 60 days post-anthesis (DPA) are shown in Figure 1A. There was no significant difference in fruit size between the two varieties throughout the developmental stages; however, WP190 exhibited a slightly elongated fruit shape with a smaller cross-sectional diameter compared to ZH108. Similarly, the fruit color characteristics showed that the fruits of WP190 and ZH108 were dark green and light green (whitish), respectively, at 5, 10, 15 and 20 DPA. In both materials, the initial change in fruit color from green to red was observed at 30 DPA, but the change was more pronounced and occurred rapidly in WP190 fruits at 40 DPA, with WP190 fruits exhibiting a redder color at 50 DPA and 60 DPA compared to ZH108. This suggests that 40 DPA is a critical stage for tomato fruit color and metabolite changes. Therefore, to determine the mechanism, the metabolomes of the two varieties were analyzed at six fruit developmental stages using liquid chromatography–tandem mass spectrometry (LC-MS/MS). A total of 68 carotenoids were identified, of which 29 exhibited significant differences in content between the fruits of two tomato varieties at different stages. The carotenoid levels of each sample are shown in the Appendix A. Heatmap analysis revealed that the 29 carotenoids exhibited differential accumulation levels between the two varieties during development and ripening. Of these, 24 carotenoids (82.76%) accumulated more in ripe fruits, including lycopene; (E/Z)-lycopene; and α-, β- and γ-carotenoids. Interestingly, the level of these carotenoids was much higher in WP190 fruits at 50 and 60 DPA in comparison to that in ZH108 fruits (Figure 1B). In addition, statistical analysis was conducted to determine the significance of differences in carotenoid content between the fruits of the two tomato varieties at each stage (Appendix A). The results showed that six carotenoids exhibited significant differences in content between WP190 and ZH108 fruits at 10 DPA, with all six being higher in WP190 fruits. Similarly, a total of 12, 11 and 11 carotenoids showed significant differences in content in fruits at 20, 30 and 40 DPA, respectively. In addition, 24 carotenoids were significantly higher in WP190 fruits compared to ZH108 fruits at 50 and 60 DPA.

### 2.2. An Overview of the Transcriptomic Data in Tomato Fruit

Transcriptomic sequencing data for the 36 samples were obtained, and raw reads were filtered to remove low-quality reads. A total of 6 GB of clean reads was obtained for each sample, with a Q20 value of more than 97% and a Q30 value of more than 92%. The distribution of GC content ranged from 41.72% to 46.36%, indicating high transcriptome quality (Appendix A). All clean reads were mapped to the SGN (Sol Genomics Network, Sol: 4.0) reference genome, and the percentage of mapped reads for each sample is presented in the Appendix A. Principal component analysis (PCA) was employed to visualize and evaluate the overall differences in gene expression among fruits of two varieties at different stages. According to the PCA, the three replicates of each sample type were well separated from each other, indicating a high consistency and quality of the data. PC1 and PC2 explained 41.66% of the variation observed, and the first PC (29.11%) separated all samples according to the developmental stages. In addition, the results showed that the WP190 and ZH108 had similar PC1 values at 10, 20 and 30 DPA. However, variations between the two varieties became apparent at 40 DPA and were more significant at 50 and 60 DPA (Figure 2A), indicating that the transcriptional variation between the two tomato varieties was primarily observed during the fruit ripening stages (i.e., 40, 50 and 60 DPA). To further analyze the transcript levels of the two tomato varieties at different stages of fruiting, the raw count data were subjected to differential expression analysis using DESeq2 to identify the differentially expressed genes (DEGs) (Appendix A). The number of DEGs was counted in two adjacent stages of two varieties. The highest number of differential genes appeared in both varieties at 40 DPA versus 50 DPA. This suggests that 40 and 50 DPA are key stages in the transcriptional regulation of tomato fruit ripening. In addition, the number of DEGs between two tomato varieties at the same stage was also quantified, and the results demonstrated that the highest number of DEGs was observed at 50 DPA (Appendix A). The transcriptome Venn diagram demonstrated that the DEGs were most abundant at 50 DPA (3946), with approximately 45.8% and 12.9% common or overlapping with those of 60 and 40 DPA, respectively (Figure 2B). Further, we performed a KEGG enrichment analysis on all the overlapped DEGs in two tomato varieties at the ripening stages (40, 50 and 60 DPA) shown in Figure 2B and found that these DEGs were associated with starch and sucrose metabolism, carotenoid biosynthesis, carbon metabolism, glycolysis/gluconeogenesis and plant hormone signals (Figure 2C).

### 2.3. Gene Screening Using WGCNA Analysis

To analyze the correlation between carotenoid metabolism and transcriptome levels, we constructed a weighted correlation network (WGCNA) to classify co-expressed gene modules. A thresholding power of 18, which best fits the scale-free topological index, was selected, and the dynamic analyses were merged to show 20 modules. The correlation coefficients between the modules and carotenoid content varied widely, ranging from −0.88 to 0.93 (Figure 3A). Among these modules, the brown and pink modules exhibited the highest positive correlation with most carotenoid metabolism levels (GS values greater than 0.5). The heatmap of the brown module showed that the expression level of these genes was upregulated with fruit development and was higher in WP190 ripe fruits compared to those of ZH108. Likewise, a similar trend was observed for the genes in the pink module, with the expression of these genes being considerably higher in WP190 fruits at later stages of development than in the early stages. Furthermore, the expression levels of the genes in the pink module peaked at 50 DPA and exhibited a decline in expression at 60 DPA (Appendix A). The overall results indicated that the genes of the brown and pink modules were significantly associated with carotenoid accumulation and fruit ripening processes in tomato. Further, KEGG pathway analysis of the brown and pink modules demonstrated that the majority of the genes were enriched in plant primary and secondary metabolic biosynthesis pathways (Figure 3B and Appendix A). Moreover, some of the genes in the pink module were also found to be enriched in metabolic pathways related to fruit ripening, including the cysteine and methionine metabolism (ethylene synthesis precursor) and the carotenoid synthesis pathway. The gene co-expression network of the pink and brown modules revealed the presence of 5 and 11 carotenoid synthesis pathway-related genes, respectively. These included PSY, PDS, ZDS and ZCIS. Additionally, 12 and 7 transcription factors were found to be co-expressed with these genes, comprising MYB, WRKY, NAC and bHLH transcription factors. In particular, a number of ERF transcription factors were co-expressed with carotenoid synthesis pathway genes in the two modules. Further analysis of the co-expression network of genes related to ethylene biosynthesis and signal transduction pathways within the modules revealed that the genes encoding the key enzymes for ethylene biosynthesis, ASC and ACO, were presented in both modules and were co-expressed with a variety of transcription factors, including ERF (Appendix A). Therefore, it can be concluded that the genes in the pink module may be associated with carotenoid accumulation, color development and fruit ripening in tomato.

### 2.4. Expression Patterns and Validation of Genes in the Carotenoid Biosynthesis and Signal Transduction Pathway

To determine the expression profiling of carotenoid biosynthesis pathway genes, RNA-seq was performed on tomato fruit at six developmental stages. The results showed that transcript levels of the 31 genes involved in the carotenoid biosynthesis were differentially expressed (The gene IDs are listed in Appendix A). Notably, most of these genes, particularly those involved in catalyzing the synthesis of lycopene and its precursors, such as *SlPSY*, *SlPDS*, *SlZCIS* and *SlCRTSO2* exhibited upregulation. In contrast, certain genes, like *SlLCYB1*, *SlLCYB2* and *SlLCYE*, which are involved in lycopene catabolism, exhibited a gradual decline in transcript levels with fruit ripening. Additionally, the expression levels of α-carotene and β-carotene catabolic pathways-related genes were also upregulated, but the trend was relatively slow (Figure 4A). Furthermore, the expression profiles of lycopene synthesis-related genes including *SlPSY2*, *SlZCIS*, *SlPDS*, *SlZDS* and *SlCRTSO2* were validated in the fruits of the two varieties using qRT-PCR. The results revealed that all these genes were highly expressed at 50 and 60 DPA (ripe fruits) in both varieties, with the majority reaching their maximum values at 50 DPA. However, significant differences were observed in the transcript levels of these genes between the two selected materials and ripening stages. The ripe fruits of WP190 displayed significantly higher expression of these genes than those of ZH108. Conversely, the *SlLCYB2* gene, which encodes a key enzyme in lycopene catabolism, was downregulated in the ripe fruits, and its expression was found to be lower in WP190 fruits compared to ZH108 (Figure 4B). To verify the reliability of RNA-seq, we plotted the relative expression data in log2FC, standardized them uniformly with RNA-seq data and then performed Pearson’s correlation analysis (Appendix A). The closer the R^2^ value to 1, the stronger the correlation between the two groups of data, and the R^2^ values between qRT-PCR data for six genes and their transcriptome data were higher than 0.8, indicating a strong correlation.

### 2.5. Expression Patterns and Validation of Genes in the Ethylene Biosynthesis and Signal Transduction Pathway

In the current study, we selected and analyzed 18 genes involved in ethylene biosynthesis with various transcript levels at different stages. Several of these genes belong to the same gene family, including, L-methionine (Met), S-adenosyl-L-methionine (SAM), 1-aminocyclopropane-1-carboxylate (ACC) and ACC oxidase (ACO) (the gene IDs are listed in Appendix A). The results indicated that certain ethylene biosynthesis-related genes were upregulated and expressed during fruit development. Moreover, the expression of specific genes within the same family was significantly higher at the fruit ripening stage. The ACS family comprises seven genes, three of which (*SlACS**1*, *SlACS4* and *SlACS7*) exhibited significantly upregulated expression during fruit ripening. Similarly, the ACO family, containing seven genes, had four SlACOs (*SlACO1*, *SlACO3*, *SlACO4* and *SlACO7*) with upregulated expression. In addition, a comparison of the two varieties revealed that certain genes, especially *ACS* (*SlACS**1* and *SlACS7*) and *ACO* (*SlACO3* and *SlACO4*), exhibited significantly higher transcript levels in the fruits of WP190 than in those of ZH108. A subsequent analysis of the expression profiles of genes related to the ethylene signal transduction pathway identified 37 DEGs belonging to six different families. The results indicated that different genes within the same family showed both up- and downregulation at various developmental stages. For example, six members of the ethylene response factor gene family were detected, with three being upregulated (*Sl**ERF1, SlERF5* and *SlERF6*), two being downregulated (*SlERF2* and *SlERF3*) and one exhibiting the highest expression during the mid-development stage of the fruits and lower expression at the green-fruit and ripening stages (*SlERF4*). This suggests the complexity of ethylene signal transduction during the physiological process of fruit ripening (Figure 5A). The expression of the six hub genes related to ethylene biosynthesis and responsiveness was verified by qRT-PCR analysis. Among the ethylene biosynthesis-related genes, *SlACO1*, *SlACO4*, *SlACS1* and *SlACS4* were highly expressed in both varieties at 50 and 60 DPA. However, the expression levels of these genes reached a maximum at 50 DPA in WP190 fruit and 60 DPA in ZH108 fruit. Similarly, the two ethylene-responsive genes *SlEIN3* and *SlERF1* showed higher expression in ripe fruits, and both genes exhibited a peak in expression at 60 DPA in ZH108. In the case of WP190, the expression trend of *SlERF1* was consistent, while *SlEIN3* expression peaked at 50 DPA and then decreased at 60 DPA. Nevertheless, all six genes related to ethylene synthesis and signal transduction exhibited significant differences in their expression levels between the two varieties (Figure 5B). In addition, Pearson’s correlation analysis was performed between qRT-PCR and RNA-seq data (Appendix A). The correlation between the two was found to be high, with the exception of differences in the peak expression of *SlACO1*.

## 3. Discussion

The ripening of tomatoes is accompanied by a change in fruit color from green to orange and finally to red. This color change is associated with the synthesis and accumulation of carotenoids during ripening [24]. In our field observations, we compared two varieties of cultivated tomatoes, WP190 from Israel and the native ZH108 tomato, and found that WP190 matured earlier and produced redder fruits than ZH108. The metabolomic results demonstrated that WP190 accumulated most of the carotenoids, including lycopene; (E/Z)-lycopene; and α-, β- and γ-carotenoids, and the content of these carotenoids was considerably higher in ripe fruits of WP190 than in fruits of ZH108 (Figure 1). The red and orange colors of tomato fruits are primarily due to differences in carotenoid content [25]. The accumulation of carotenoids is regulated by the expression of genes encoding enzymes in their pathways. Some of the genes encoding key enzymes in the carotenoid synthesis pathway include PSY, PDS, ZDS, CRTISO, ZISO and LCYE or LCYB. The up- and downregulated expression of these genes can regulate carotenoid content by catalyzing carotenoid synthesis and catabolism. The molecular mechanism of zucchini (*Cucurbita pepo* L.) fruit ripening and coloration has been revealed by the analysis of integrating carotenoid metabolites and the transcriptome [26]. In this study, based on phenotypic observation, the transcriptomic and metabolomic data were used together to construct a co-expression network for WGCNA analysis to analyze the correlation between transcript levels and carotenoid metabolic levels. We identified certain co-expression modules of genes related to carotenoid and ethylene biosynthesis and signaling between the fruits of two tomato varieties at different developmental stages.

Tomato fruits are rich in various carotenoids, including lycopene, β-carotene and lutein, and ripe red fruits are mainly associated with lycopene accumulation. The regulatory mechanism of carotenoid metabolism during fruit ripening is systematic and complicated [27]. Transcriptional regulation of structural genes is the primary mechanism for carotenoid accumulation, and the process is linked with a combination of phytohormonal and environmental signals [28,29,30]. Transcriptome analysis revealed that the major transcriptional differences occurred at the breaker stages (40–50 DPA), when WP190 fruits began to turn light red and ZH108 fruits began to turn light orange. In addition, the two tomato varieties exhibited greater differences at the same stage at 40, 50 and 60 DPA. These DEGs of the two varieties at 40, 50 and 60 DPA were enriched in various metabolic pathways of plant development, mainly fruit ripening, phytohormone signaling and carotenoid biosynthesis pathways (Figure 2). Furthermore, transcriptomic and metabolomic data were utilized collectively to construct a co-expression network for WGCNA analysis. The genes in the identified pink module were found to be significantly associated with the carotenoid and ethylene biosynthesis and signal transduction pathways, suggesting that these genes may have similar functions in the regulation of fruit ripening and color transfer (Figure 3).

Phytoene is the initial intermediate of the carotenoid pathway, and its production is considered as the primary rate-determining step in the regulation of metabolic fluxes within this pathway [31]. Previously, it was reported that three PSY gene members regulate carotenoid biosynthesis in distinct tomato tissues [6]. In addition, the expression of *SlPSY1* increased as the tomato fruit ripened, resulting in the accumulation of carotenoids and lycopene, which gradually transformed the fruit from green to orange or red [32,33,34]. Similarly, *SlPSY2*, a highly homologous gene of *SlPSY1*, exhibited significantly higher expression in all the tissues compared to *SlPSY1* [32,33,34]. However, *SlPSY2* is mainly associated with signaling mechanisms in underground plant parts and expressed prominently in root tissues [35,36]. In our study, we observed that *SlPSY1* and *SlPSY2* expression levels were upregulated during fruit ripening periods. Interestingly, *SlPSY2* was expressed differently between the two varieties and showed maximum expression at 50 and 60 DPA in WP190 and ZH108, respectively, but significantly higher expression in the former (Figure 4A). The expression pattern of *SlPSY2* was consistent with the trend of phenotypic changes and phytoene content in both varieties. Similarly, the high expression of PSY may have triggered the synthesis of phytoene and the reddening of fruits. The red color formation at the breaker stage is a characteristic feature of high lycopene contents in tomato fruit, and lycopene accumulation is significantly induced by the expression of many genes, including PDS, ZDS, Z-ISO and CrtISO, which are situated upstream in the lycopene biosynthesis pathway [37,38,39,40]. In our study, we observed that the expression levels of *SlPDS*, *SlZDS*, *SlZCIS* and *SlCRTISO2* were significantly elevated in the red ripe fruits of WP190 compared to other developmental stages and orange ripe fruits of ZH108 at the same stage (Figure 4B). Similarly, the downregulation of the lycopene catabolism genes during the ripening stages induced lycopene accumulation and enhanced the dark red color of the fruits [41]. LCYB is a key enzyme involved in the conversion of lycopene to carotene, and it has been reported that the activity and expression of LCYB are downregulated during fruit ripening [25]. In the current study, the expression of *SlLCYB2* was found to be higher in fruits at 30 and 40 DPA, to start decreasing at 50 DPA, and to be lowest at 60 DPA. In comparison to ZH108, WP190 displayed a relatively low expression level of *SlLCYB2* at 50 and 60 DPA, which may be one of the contributing factors to the higher lycopene accumulation observed in ripe WP190 fruits.

Tomato, as a climacteric fruit, displays an increase in respiration and ethylene synthesis upon the initiation of ripening. Ethylene is essential for fruit ripening, and blocking either the biosynthesis or perception of ethylene can prevent fruit ripening [42,43]. Ethylene mediates ripening and activates the expression of related genes to promote carotenoid biosynthesis in cultivated tomatoes [44]. Based on the carotenoid content of tomato as an indicator of fruit ripening, combined with fruit transcriptome data from different stages, it was found that the expression of genes related to the ethylene biosynthesis and signaling pathways had a corresponding trend in the fruit ripening process of both tomato varieties. Differences in the transcript levels of these genes determined the interaction between fruit ripening and ethylene biosynthesis in tomato. ACS and ACO are the two rate-limiting key enzymes in ethylene biosynthesis [16]. There are 11 ACS genes in tomato, and they are differentially expressed at different stages of development and ripening [45]. Differential expression of ACS genes is induced by ethylene-mediated autocatalysis during tomato fruit development and ripening, and their expression level varies significantly between fruit development and ripening stages [46]. ACS genes exhibited moderate expression during fruit development and were induced by endogenous mechanisms during ripening, with high expression of *ACS1*, *ACS4* and *ACS6* in tomato [47]. Previous studies have indicated a positive regulatory relationship between tomato fruit ripening and *ACS1* expression; however, the antisense expression of the *ACS1* gene has been shown to result in a delay in tomato fruit ripening [48]. In addition, the high expression of ACS1 during fruit ripening may be regulated by transcription factors such as RIN [49] and TAGL1 [50]. We analyzed and validated the expression profiles of four ACS genes and three ACO genes that were significantly upregulated in ripening fruit (Figure 5A). In WP190, the expression of *SlACS1*, *SlACS4*, *SlACO1* and *SlACO4* was found to be upregulated with fruit ripening, with a peak observed at 50 DPA, followed by a relative decrease in red-ripened fruit (60 DPA). In ZH108 fruits, the expression of *SlACS1*, *SlACS4* and *SlACO4* reached a maximum only at 60 DPA. In a comparison of the two varieties, the transcription levels of *SlACS1* and *SlACO4* were significantly higher in WP190 than in ZH108 fruits of the same stage (Figure 5B). This suggests that ASC- and ACO-regulated ethylene biosynthesis is the key factor responsible for the observed differences in ripening between the two varieties of tomato.

Ethylene signal transduction regulates fruit development and ripening [51,52], and the ethylene sensing and downstream signaling cascades are well established in Arabidopsis [53]. Compared to Arabidopsis, the tomato genome encodes more ethylene signal transduction components, indicating greater flexibility of ethylene signaling in tomato [42]. Comprehensive genetic analyses of the tomato genome uncovered a critical function of ethylene in fruit growth and ripening, as loss of function of the key ethylene signaling component ETHYLENE INSENSITIVE 2 (*SlEIN2*) resulted in the inhibition of fruit ripening and size [14]. Members of the EIN3-like (EIL) family have been reported to play an important role in fruit development, and the antisense lines of *SlEIL1*, *SlEIL2* and *SlEIL3* homologs to EIN in tomato produced yellow-colored fruits [54]. In addition, the tomato ERF transcription factors are involved in the ethylene signaling pathway and actively participate in fruit development and ripening. The *SlERF6* member of the ERF family has been shown to regulate carotenoid synthesis in tomato fruit [55]. Similarly, *SlERF.B3* controls fruit ripening by regulating ethylene synthesis and carotenoid accumulation [56]. Expression profiling of ERFs in wild-type and ripening-impaired tomato mutants (never-ripe [Nr], ripening-inhibitor [rin] and non-ripening [nor]) revealed that different ERFs have contrasting roles in fruit ripening [57]. In the present study, several ethylene-responsive genes were found to be differentially expressed during fruit development in two varieties of tomato. *SlEIN3* expression was higher in the WP190 fruit at 50 DPA, while in the ZH108 fruit, this was observed at 60 DPA. The transcript level of *SlERF1* was consistently increased during fruit ripening in both varieties; however, the WP190 fruits showed significantly higher expression compared to ZH108. Taken together, the expression trends of ethylene synthesis and responsive genes were consistent with the degree of tomato fruit ripening, thereby revealing the key genes regulating fruit ripening and their potential functions.

Tomato fruit ripening is accompanied by the accumulation of ethylene and carotenoids. This study identified two key pathways related to the ripening and coloring process of tomato fruit, as revealed by the transcriptomes and metabolomes of the fruit. Furthermore, the regulatory patterns of genes related to fruit development and color formation at different ripening stages were verified, and key carotenoid biosynthesis pathway genes that promote high lycopene in WP190 and exhibit unique expression patterns in lycopene biosynthesis compared to ZH108 were identified. Overall, this study contributes to our understanding of the regulatory mechanisms of tomato fruit ripening.

## 4. Materials and Methods

### 4.1. Plant Material and Sample Preparation

Tomato seeds (*S. lycopersicum*) of two varieties, WP190 (W) and ZH108 (Z), were collected from the Key Laboratory of Genome Research and Genetic Improvement of Xinjiang Characteristic Fruits and Vegetables. The seeds were sown in pots containing a mixture of soil, vermiculite and peat. The young seedlings were maintained in an artificial greenhouse with controlled environmental conditions (temperature of approximately 25 ± 1 °C, relative humidity of 50 ± 10%, a photoperiod of 16 h/8 h and three irrigations of 1/2 Hoagland’s nutrient solution). Six-week-old seedlings of both varieties were transplanted to the field and kept under natural growing conditions with sufficient sunlight. The fruit samples were collected at different developmental stages, i.e., 5, 10, 15, 20, 30, 40, 50 and 60 days post-anthesis (DPA). The stages were as follows: 10 and 20 DPA (green fruit), 30 DPA (mature green), 40 DPA (breaker stage), 50 DPA (light red fruit of WP190 and light orange fruit of ZH108) and 60 DPA (red fruit of WP190 and orange fruit of ZH108). The samples of whole fruit (including the skin and flesh) were immediately frozen in liquid nitrogen and stored at −80 °C for metabolome profiling, mRNA sequencing and qRT-PCR validation.

### 4.2. Carotenoid Identification and Quantification

For the quantification of carotenoids, the freeze-preserved lyophilized samples were ground using zirconia beads (30 Hz, 1 min), and the resulting powder (50 mg) was dissolved in 0.5 mL of the extraction solution (n-hexane/acetone/ethanol (1:1:1, *v*/*v*/*v*) and 0.01% butylated hydroxytoluene). Subsequently, the homogenate was incubated and vortexed for 20 min at room temperature and centrifuged at 12,000× *g* for 5 min at 4 °C. The supernatant was collected and dissolved in 100 μL of a methanol/methyl tert-butyl ether mixture (1:1, *v*/*v*) in order to facilitate filtration (PTFE, 0.22 μm; Anpel, Shanghai, China). The solution was then stored in a brown injection vial for subsequent LC-MS/MS analysis. The carotenoid identification and quantification analyses were performed by Metware Biotechnology Co., Ltd. (Wuhan, China), utilizing an LC-APCI-MS/MS system. The detection of the contents of carotenoids was achieved using UPLC (ExionLC AD) and a tandem mass spectrometer (MS/MS) (QTRAP 6500+, N). Chromatographic separations were conducted on a YMC C30 (3 µm, 2 mm × 100 mm) column maintained at 28 °C. The flow rate was set at 0.8 mL/min, and the mobile phase consisted of two different solvents, acetonitrile/methanol (1:3, *v*/*v*), 0.01% butylated hydroxytoluene (BHT) and 0.1% formic acid solution (solvent A) and methyl tert-butyl ether and 0.01% BHT solution (solvent B). The gradient programs for the solvent A/solvent B solutions were as follows: 100:0 *v/v* at 0 min, 100:0 *v*/*v* at 3 min, 30:70 *v*/*v* at 5 min, 5:95 *v*/*v* at 9 min, 100:0 *v*/*v* at 10 min, 100:0 *v*/*v* at 11 min, and a total of 2 μL sample was used for the quantification process. The quantification of the carotenoids was achieved using calibration curves for 12 standards. The standard curve equation and correlation coefficients for the detected substances are presented in the Appendix A.

### 4.3. RNA-Seq Analysis

Total RNA was extracted from the fruits of two tomato varieties at six different stages, with three biological replicates. RNA was fragmented into short fragments by the addition of fragmentation buffer, and double-stranded cDNA was synthesized using the short-fragmented RNA as a template. Purification was then performed using AMPure XP beads. The purified double-stranded cDNA was then subjected to end repair, A-tailing and ligation to sequencing junctions. Fragment size selection was performed using AMPure XP beads, and finally PCR enrichment was employed to obtain the final cDNA library. A Qubit2.0 was used for preliminary quantification, while an Agilent 2100 was utilized to detect the insert size of the library. The sequencing was conducted on an Illumina HiSeq4000 platform, the adapters and low-quality sequences were removed using Fastp (v0.19.3, https://github.com/OpenGene/fastp; accessed on 20 September 2023) with the default parameters, and the remaining clean reads were then mapped to the tomato genome (version SL4.0) using HISAT2 [58]. HTSeq v0.6.125 was used to count the read numbers mapped to each gene, and FPKM was used for the quantification of genes and transcripts at the gene/transcript level. The raw count data were subjected to differential expression analysis using DESeq2 software. The false discovery rate (FDR) was obtained by correcting the probability of hypothesis testing (*p* value) for multiple hypothesis testing using the Benjamini–Hochberg approach. Genes exhibiting a |log2Fold Change| ≥ 1 and an FDR < 0.05 were defined as differentially expressed genes (DEGs). Principal component analysis (PCA) was employed to identify associations in the transcriptomic data between samples. The results were analyzed and visualized using R Studio software (https://www.rstudio.com/; accessed on 25 March 2024) with the FactoMineR package.

### 4.4. Co-Expression Network Analysis for Construction of Modules

For co-expression network analysis, the weighted gene co-expression network analysis (WGCNA) package was used. To identify genes associated with carotenoid synthesis, differentially expressed genes (DEGs) and differentially accumulated carotenoids (DACs) were detected in the six developmental stages of fruit for integrative analysis. The DEGs with DAC Pearson correlation coefficient (PCC) ≥0.90 or ≤−0.90 were selected for subsequent WGCNA with the default parameters and subjected to KEGG enrichment analysis. The co-expression network was visualized using the free software Cytoscape (v3.10.2).

### 4.5. RNA Extraction and Quantitative Real-Time PCR (qRT-PCR)

Total RNA was extracted from the selected samples using the Polysaccharide Polyphenol Plant Total RNA Extraction Kit (DP441, Tiangen, Beijing, China) in accordance with the manufacturer’s instructions. The first strand of cDNA was synthesized using the 5×All-In-One RT Master Mix, AccuRT Genomic DNA Removal Kit (G492, ABM, Vancouver, BA, Canada). qRT-PCR specific primers were designed from the coding region of the selected genes using DNAMAN 8 software, and all the primers are listed in the Appendix A. qRT-PCR was implemented using ChamQ Universal SYBR qPCR Master Mix (Q711, Vazyme, Nanjing, China) on a LightCycler^®^ real-time fluorescent quantitative PCR system (Roche, Basel, Switzerland). Slactin (Solyc03g078400.2) was used as an internal reference control. For each sample, three technical replicates were performed to calculate the mean cycle threshold (Ct) value. The relative expression levels were calculated using the 2^−ΔΔCt^ method.

### 4.6. Statistical Analysis

All data were of qRT-PCR analyzed using GraphPad Prism 8.0 (GraphPad Software, San Diego, CA, USA). Significant differences between the groups were analyzed with one-way and two-way ANOVA. The differences were compared with Tukey’s method for multiple comparisons at a significance level of 0.05, 0.01 or 0.001.

## 5. Conclusions

In this study, we compared the metabolomes and transcriptomes of two tomato varieties characterized by different fruit colors. The content of carotenoids increased with fruit ripening, and the levels of carotenoid compounds, including lycopene; (E/Z)-lycopene; and α-, β- and γ-carotenoids, were significantly higher in WP190 than in ZH108, particularly during the fruit ripening stages (50 and 60 DPA). Transcriptome analysis showed that most of the differentially expressed genes between the two varieties were predominantly abundant in fruits at 50 DPA. In WGCNA, we identified two modules that were positively associated with carotenoid accumulation, and genes related to carotenoid and ethylene biosynthesis and signaling pathways were also enriched. An analysis of the expression profiles of genes associated with these pathways revealed that the expression of genes involved in lycopene biosynthesis was upregulated, while the expression of a key gene involved in lycopene catabolism was downregulated. This resulted in a significant increase in lycopene content in red ripe fruits. Similarly, genes related to the ethylene biosynthesis and signaling pathway, including *SlACS1*, *SlACS4*, *SlACO2*, *SlACO4*, *SlEIN3* and *SlERF1*, were upregulated during fruit ripening, with significant differences observed between the two varieties. The results indicate that tomato fruit ripening is regulated by genes associated with the carotenoid and ethylene biosynthesis and signaling pathways, and differential expression of these genes distinguishes the two varieties in terms of carotenoid content accumulation and fruit color formation. This study not only provides new insights into the synthesis and accumulation of carotenoids in tomato fruit but also lays a solid biological foundation for the selection and breeding of high-lycopene tomato varieties.

## Figures and Tables

**Figure 1 ijms-25-06493-f001:**
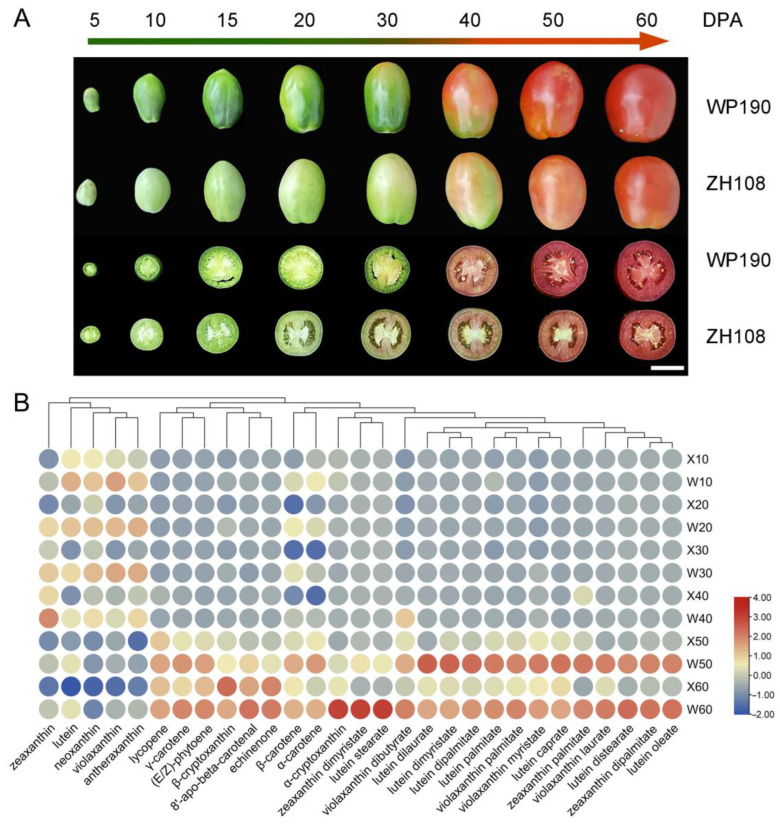
The fruit phenotypes and carotenoid metabolomics analysis of six development stages of tomato fruit. Morphology of WP190 and ZH108 fruits at six developmental stages, scale bar = 1 cm (**A**). Heatmap of carotenoid metabolism in tomato fruits at different developmental stages (**B**).

**Figure 2 ijms-25-06493-f002:**
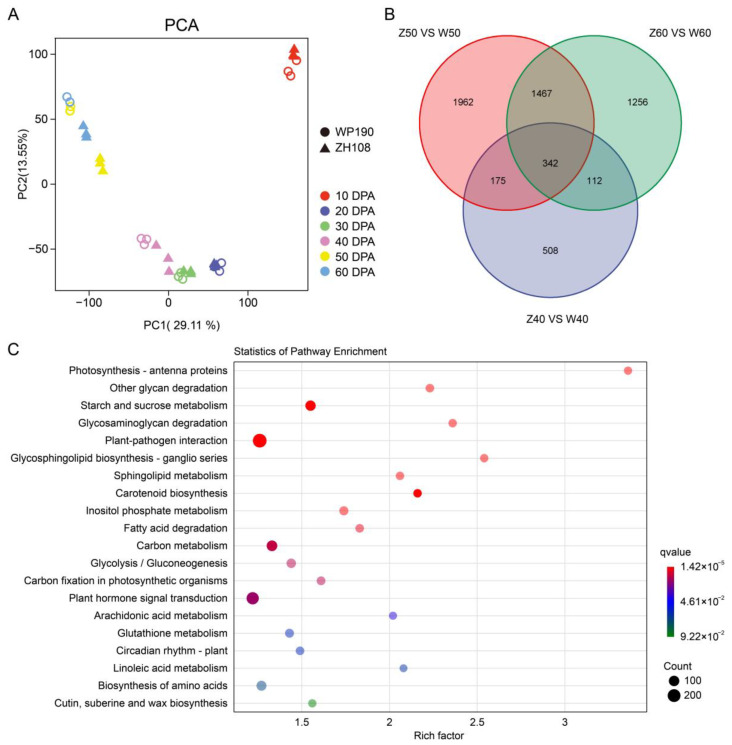
Variability of transcript levels between two tomato varieties. (**A**) PCA results of the transcriptome data. (**B**) Overlap of DEGs in two varieties compared with each other at 40, 50 and 60 DPA; the numbers indicate the numbers of up- and downregulated genes. (**C**) The first twenty KEGG pathways enriched in the DEGs of two tomato varieties at 40, 50 and 60 DPA (in (**B**)).

**Figure 3 ijms-25-06493-f003:**
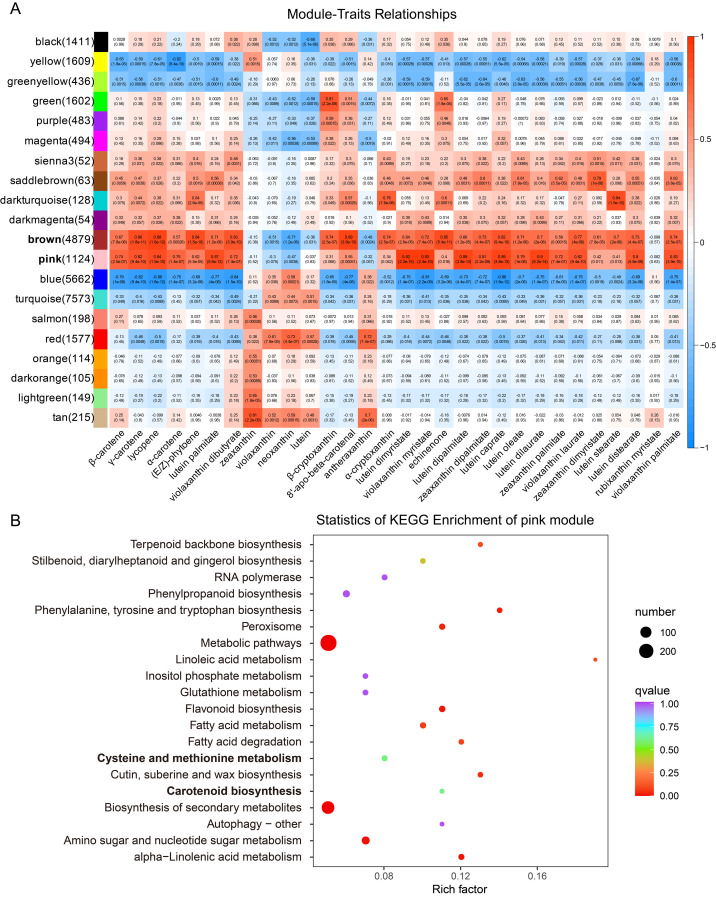
Module–carotenoid association analysis. (**A**) Heatmap showing the correlation between modules and carotenoids. The GS value between a given module and carotenoid is indicated by the color of the cell and the text inside the cell (the upper number is the value, and the lower number is the *p* value). Red and blue indicate positive and negative correlations, respectively. (**B**) KEGG pathway enrichment analysis of pink module.

**Figure 4 ijms-25-06493-f004:**
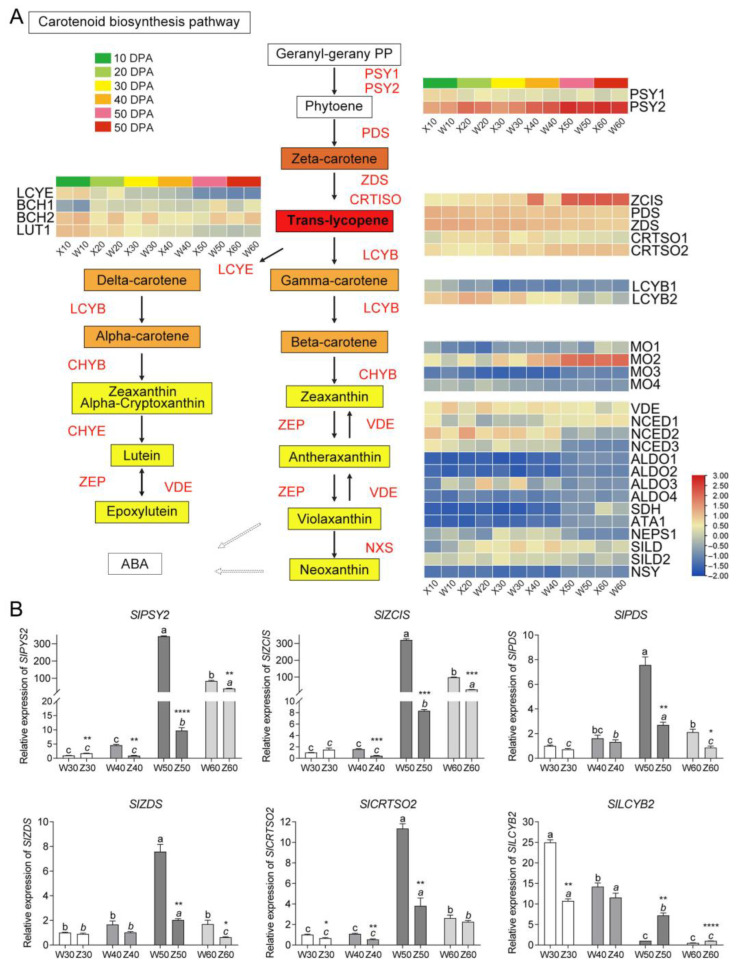
Expression patterns of genes in the carotenoid biosynthesis and signal transduction pathway. (**A**) Transcript profiles for genes in the carotenoid biosynthesis and signal transduction pathway. Grids with a color scale from blue to yellow to red represent the gene expression of the DEGs from low to medium to high. PSY, phytoene synthase; ZCIS, 15-cis-zeta-carotene isomerase; PDS, phytoene dehydrogenase; ZDS, zeta-carotene desaturase; CRTSO, zeta-carotene desaturase; LCYE, zeta-carotene desaturase; ABA2, zeaxanthin epoxidase2; LCYB, beta-lycopene cyclase; MO, monooxygenase; VDE, volaxanthin de-epoxidase; NCED, 9-cis-epoxycarotenoid dioxygenase; ALDO3, abscisic-aldehyde oxidase; SDH, secoisolariciresinol dehydrogenase; ATA1, short-chain dehydrogenase reductase1; NEPS1, nepetalactol synthase1; SILD, secoisolariciresinol dehydrogenase; NSY, neoxanthin synthase; BCH, beta-carotene hydroxylase; ABAH, abscisic acid 8′-hydroxylase; LUT, protein lutein deficient; D27, protein DWARF-27 (beta-carotene isomerase). (**B**) qRT-PCR data of carotenoid biosynthesis genes. Different lowercase letters above columns indicate significant differences (*p* < 0.05). *, **, *** and **** represent *p* < 0.05, *p* < 0.01, *p* < 0.001 and *p* < 0.0001 between the two varieties of tomato fruits at the same stage.

**Figure 5 ijms-25-06493-f005:**
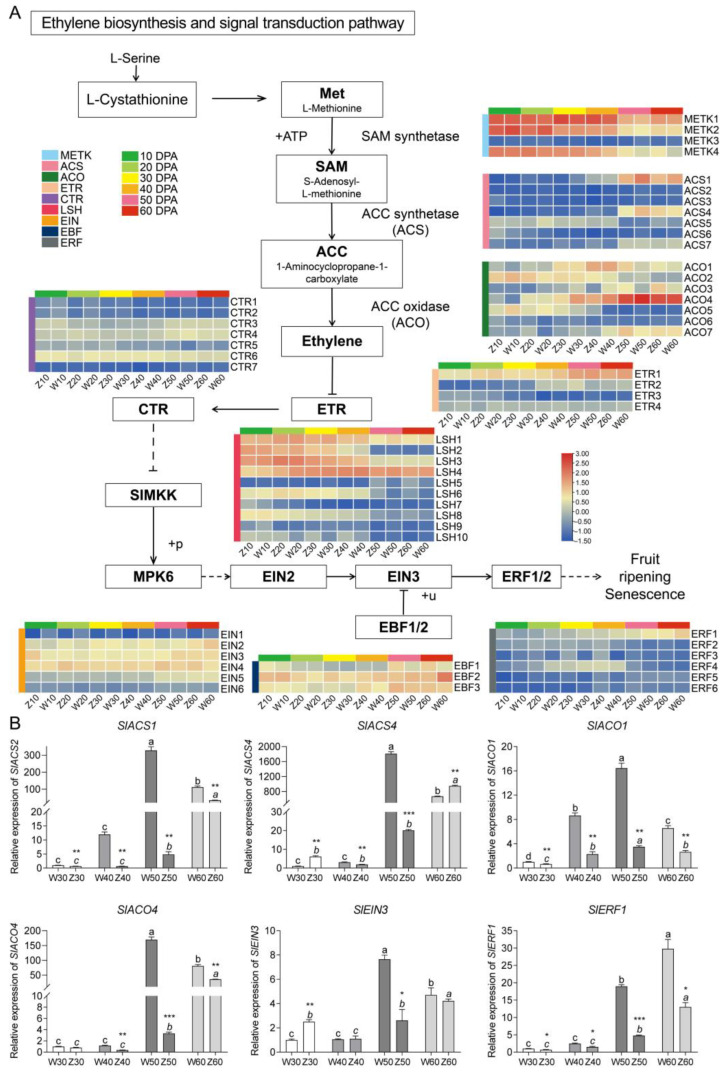
Expression patterns of genes in the ethylene biosynthesis and signal transduction pathway. (**A**) Transcript profiles for genes in the ethylene biosynthesis and signal transduction pathway. Grids with a color scale from blue to yellow to red represent the gene expression of the DEGs from low to medium to high. Met, L-methionine; SAM, S-adenosyl-L-methionine; ACC, 1-aminocyclopropane-1-carboxylate; ACO, ACC oxidase; ETR, ethylene receptor; CTR, constitutive triple response; MKK, mitogen-activated protein kinase kinase; MPK, mitogen-activated protein kinase; LSH, protein light-dependent short hypocotyls; EIN, ethylene insensitive; EBF, EIN3-binding F-box; ERF, ethylene-responsive transcription factor. (**B**) qRT-PCR data of ethylene biosynthesis and response genes. Different lowercase letters above the column indicate significant differences (*p* < 0.05). *, ** and *** represent *p* < 0.05, *p* < 0.01 and *p* < 0.001 between the two varieties of tomato fruits at the same stage.

## Data Availability

All relevant data are presented within the paper and its Appendix A. Transcriptome sequencing data are publicly available at NCBI (Accession number: BioProject ID: PRJNA1115050).

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
