# Peer review of "Integrative Analysis of Metabolome and Transcriptome of Carotenoid Biosynthesis Reveals the Mechanism of Fruit Color Change in Tomato (*Solanum lycopersicum*)"

_ijms, 2024, doi:10.3390/ijms25126493_

Round 1
Reviewer 1 Report
Comments and Suggestions for Authors
Tomato fruit ripening involves carotenoid accumulation and color changes. To understand this process, authors conducted transcriptomic and metabolomic analyses on red-fruited (WP190) and orange-fruited (ZH108) tomatoes. They identified 29 carotenoid compounds in the fruits at six different stages, with most carotenoids increasing significantly as the fruit ripened. Higher levels of lycopene and other carotenoids were found in WP190 at 50 and 60 days post anthesis (DPA). Transcriptome analysis showed the highest number of differentially expressed genes (DEGs) at 50 DPA, and a gene module related to carotenoid content was identified. qRT-PCR confirmed increased expression of lycopene biosynthesis genes in WP190 at 50 DPA compared to ZH108. Additionally, genes involved in ethylene biosynthesis and signaling also increased during ripening, suggesting they regulate carotenoid accumulation and fruit ripening. This study provides insights into the mechanisms of carotenoid accumulation and ethylene-related regulation in tomato ripening.
Overall, the content and purpose of the study are well-suited for publication. The experiments were conducted thoroughly, and the authors have prepared the figures effectively. The manuscript is also well-written. However, I noticed that the authors did not deposit the raw data associated with the LC-MS/MS and RNA sequencing. This is crucial for manuscripts related to metabolomics and transcriptomics. Please provide the accession numbers in the data availability section of the manuscript. Once this is addressed, the manuscript will be suitable for publication.
Author Response
Manuscript ID: ijms-3022080
Type: Research Article
Title: Integrative analysis of metabolome and transcriptome reveals the mechanism of fruit ripening in tomato (Solanum lycopersicum)
Dear Editor:
Thanks for your cooperation. We have studied the comments carefully and made corrections which we hope meet with approval.
All the correction are marked red in the marked change file.
Response to Reviewer #1
Overall, the content and purpose of the study are well-suited for publication. The experiments were conducted thoroughly, and the authors have prepared the figures effectively. The manuscript is also well-written. However, I noticed that the authors did not deposit the raw data associated with the LC-MS/MS and RNA sequencing. This is crucial for manuscripts related to metabolomics and transcriptomics. Please provide the accession numbers in the data availability section of the manuscript. Once this is addressed, the manuscript will be suitable for publication.
Response: Many thanks to the reviewer for his appreciation and suggestions. The raw transcriptome data have been uploaded to NCBI (https://www.ncbi.nlm.nih.gov/), and the transcriptome sequencing data are publicly available at NCBI (accession number: BioProject ID: PRJNA1115050). We have revised the manuscript and changed/added many statements in all sections to further improve the content of the manuscript. All changes are highlighted in the revised manuscript.

Reviewer 2 Report
Comments and Suggestions for Authors
The paper describes an interesting integrative analysis of metabolome and transcriptome data. However, the title refers to "the mechanism of fruit ripening in tomato" and the paper is focused on carotenoids and other pigments synthesis. Authors should change the title to make it more specific or add data of other fruit attributes that contribute to ripening. Then, both varieties under study appear to be very similar, just varying in color. It is possible that authors firstly attempted to make an overall study on tomato ripening, but the experimental plant material resulted to be inadequate. This is the reason why I recommend focusing the new version of the paper on pigments assessment at metabolome and transcriptome levels. Another important task for authors is to describe with more details the color module analyses to establish data associations, the use of different dates in the analyses (i.e., 15 days was not assessed in transcriptomics), and to validate the specific use of one way and two ways ANOVA. Finally, Discussion and Conclusions must be reoriented to this new proposed approach to available data.
Comments on the Quality of English LanguageEnglish is adequate but a revision is needed (as examples, "repining" in the Keywords). Also, some References are cited in a different way (for instance, "9 SUN, B.; ZHANG, F.; XUE, S.; CHANG, J.; ZHENG, A.; JIANG, M.; MIAO, H.; WANG, Q.; TANG, H. Molecular Cloning and 501 Expression...").
Author Response
Manuscript ID: ijms-3022080
Type: Research Article
Title: Integrative analysis of metabolome and transcriptome reveals the mechanism of fruit ripening in tomato (Solanum lycopersicum)
Dear Editor:
Thank you for your cooperation. We have carefully reviewed the comments carefully and have made corrections that we hope will meet with approval.
All the corrections are marked in red in the revised manuscript.
Response to Reviewer #2
The paper describes an interesting integrative analysis of metabolome and transcriptome data. However, the title refers to "the mechanism of fruit ripening in tomato" and the paper is focused on carotenoids and other pigments synthesis. Authors should change the title to make it more specific or add data of other fruit attributes that contribute to ripening.
Response: We thank the reviewers for their suggestions and acknowledge the reviewers' recommendation regarding the title of the study. The initial focus of this study was on the color differences between two tomato varieties during fruit ripening. This led to the detection of carotenoid levels in different stages of the fruits of both varieties. In a combined analysis of carotenoid metabolism and transcriptome, we found that genes related to carotenoid and ethylene biosynthesis and signaling pathways were significantly and differentially expressed in the fruits of two varieties at different developmental stages. Therefore, based on the recommendation of the reviewer we have changed the title of the study “Integrative analysis of metabolome and transcriptome of carotenoid biosynthesis reveals the mechanism of fruit color change in tomato (Solanum lycopersicum)”.
Then, both varieties under study appear to be very similar, just varying in color. It is possible that authors firstly attempted to make an overall study on tomato ripening, but the experimental plant material resulted to be inadequate. This is the reason why I recommend focusing the new version of the paper on pigments assessment at metabolome and transcriptome levels. Another important task for authors is to describe with more details the color module analyses to establish data associations, the use of different dates in the analyses (i.e., 15 days was not assessed in transcriptomics), and to validate the specific use of one way and two ways ANOVA. Finally, Discussion and Conclusions must be reoriented to this new proposed approach to available data.
Response: In this study, the WGCNA analysis was based on the correlation between metabolomic and transcriptomic data from all the samples. To further elucidate the genes associated with carotenoid and ethylene biosynthesis and signaling pathways, we conducted complementary analyses on the color modules obtained in the WGCNA analysis. The two modules exhibited a high degree of correlation with alterations in lycopene content, as well as their co-expression relationships with transcription factors. In response to the reviewers' suggestions, additional evidences have been included in the revised manuscript to illustrate how the carotenoid and ethylene biosynthesis and signaling pathway genes that play important roles in fruit development and ripening and lycopene metabolism have been explored, Supplementary Figure 3 and Supplementary Figure 4, (Line 184 - 185, 189 - 198)
The phenotypes of eight fruit stages from 5 to 60 days post-anthesis (DPA) are shown in Fig. 1A, including 5, 10, 15, 20, 30, 40, 50, 60 DPA. But the fruit color changes and carotenoid metabolism that we are primarily focused on appear to the ripening stage of the fruit, so 15 days was not assessed in transcriptomics.
Comments on the Quality of English Language
English is adequate but a revision is needed (as examples, "repining" in the Keywords). Also, some References are cited in a different way (for instance, "9 SUN, B.; ZHANG, F.; XUE, S.; CHANG, J.; ZHENG, A.; JIANG, M.; MIAO, H.; WANG, Q.; TANG, H. Molecular Cloning and 501 Expression...").
Response: Thanks to the reviewers. The manuscript undergone a comprehensive revision and all necessary corrections have been made accordance with the reviewer's suggestions (Line 30, 585 - 587).

Reviewer 3 Report
Comments and Suggestions for Authors
In the manuscript that was submitted by Qinghui Yu, Baike Wang and colleagues, entitled ‘Integrative analysis of metabolome and transcriptome reveals the mechanism of fruit ripening in tomato (Solanum lycopersicum)’, the authors study the carotenoid accumulation of two tomato genotypes during fruit ripening via integration analysis of metabolome and transcriptome.
Among the positive aspects of the manuscript are: (a) the depth of omics data analysis is satisfied, (b) the figures have a good presentation. The negative of the manuscript include; (a) the unclear purpose, (b) additional analysis is needed, especially to discover transcription factors that enable to carotenoid accumulation and ethylene signaling, (c) There is no coherence in the text, as both color development and ripening of the tomato are studied. Overall, the current manuscript is in line with the aims and purposes of the IJMS journal, however there are several issues that should be carefully addressed.
Major and minor
Abstract section
Lines 11-13; you should clearly define the aim of the study.
Lines 25-27; Provide a clear conclusion in case of carotenoid accumulation and not in fruit ripening.
Introduction section
Lines 84-86; ethylene regulates ripening and carotenoids coloration. Why not separate these? that is, to tell one story about coloring and one about ripening. My opinion is that they should not be considered together.
Results section
In fig.1, except for images of tomatoes, the authors should present and other physiological traits related to ripening such as firmness, SSC, TA, weight, and ethylene production (as study the fruit ripening via ethylene signaling).
Brown and pink modules need to be further analyzed (using cytoscape) to identify transcription factors possibly involved in the induction and accumulation of carotenoids.
Subsection 2.5.; ethylene and related genes suddenly appears (without having determined it during fruit maturation). You need to reanalyze your WGCNA data especially the modules to study again through the program cytoscape which genes are involved with ethylene biosynthesis and thus with ripening. Of course you have to determine the ethylene as well.
Generally, you don't have a transition from color development to fruit ripening; you need to create additional graphs to support this transition. The current data curation gives the impression that these are two separate studies (color development via carotenoid accumulation and fruit ripening during ethylene signaling).
Author Response
Manuscript ID: ijms-3022080
Type: Research Article
Title: Integrative analysis of metabolome and transcriptome reveals the mechanism of fruit ripening in tomato (Solanum lycopersicum)
Dear Editor:
Thank you for your cooperation. We have carefully reviewed the comments carefully and have made corrections that we hope will meet with approval.
All the corrections are marked in red in the revised manuscript.
Response to Reviewer #3
Major and minor
Abstract section:
Lines 11-13; you should clearly define the aim of the study.
Response: The aim of this study is to elucidate the regulatory mechanisms underlying carotenoid synthesis during fruit ripening and changes gave been made in the abstract section of the manuscript (Line 11-14).
Lines 25-27; Provide a clear conclusion in case of carotenoid accumulation and not in fruit ripening.
Response: Thank you to the reviewer for their suggestion. This study provides a comprehensive understanding of carotenoid accumulation and ethylene biosynthesis and signal transduction pathway regulatory mechanisms during tomato fruit development. In accordance with the reviewer's suggestion, a clear conclusion of the study has been added to the Abstract section of the revised manuscript, “Differential expression of several key genes in fruit of two tomato varieties at different stages regulates accumulation of carotenoid and leads to differences in fruit color between the two varieties of tomato” (Line 26-30).
Introduction section:
Lines 84-86; ethylene regulates ripening and carotenoids coloration. Why not separate these? that is, to tell one story about coloring and one about ripening. My opinion is that they should not be considered together.
Response: In the last paragraph of the introduction, the concept of the study is re-stated. To elucidate the underlying mechanisms for the observed differences in carotenoid metabolim and fruit color between the two tomato varieties during fruit ripening, transcriptome sequencing and metabolomic analysis were conducted at six different developmental and ripening stages. The combined data from the metabolome and transcriptome were subjected to WGCNA analysis, which allowed for the identification of co-expressed genes and transcription factors related to carotenoid metabolism at different fruit developmental stages between the two varieties. Moreever, the expression trends of genes involved in the carotenoid synthesis pathway, including SlPSY2, SlPDS, SlZCIS, SlZDS, SlCRTSO2 and SlLCYB2 were analyzed at different stages in the two tomato varieties, and the expression profiles of genes involved in ethylene biosynthesis and signaling were examined at the same time, including SlACS1, SlACS4, SlACO2, SlACO4, SlEIN3, and SlERF1. The findings of the current study reveal that the key genes involved in carotenoid and ethylene synthesis play a pivitol role in regulating ripening and coloration of tomato fruits (Line 81 - 94).
Results section:
- In fig.1, except for images of tomatoes, the authors should present and other physiological traits related to ripening such as firmness, SSC, TA, weight, and ethylene production (as study the fruit ripening via ethylene signaling).
Response: We thank the reviewers for their suggestion. In the preliminary study, we mainly focused on the trend of fruit carotenoid content by observing the phenotypic differences in fruit color between the two tomato varieties at different stages. We agree with the reviewer that firmness, SSC, TA, weight, and ethylene production are key indicators of the ripening process, and their analysis will further improve our research results. However, these physiological traits related to ripening require fresh sample material and due to the climatic limitation in Xinjiang regions, our research material is currently at the one-month-old seedling stage, and we will continue to perform this part in the subsequent in-depth study.
- Brown and pink modules need to be further analyzed (using cytoscape) to identify transcription factors possibly involved in the induction and accumulation of carotenoids. Subsection 2.5.; ethylene and related genes suddenly appears (without having determined it during fruit maturation). You need to reanalyze your WGCNA data especially the modules to study again through the program cytoscape which genes are involved with ethylene biosynthesis and thus with ripening. Of course you have to determine the ethylene as well.
Response: Thanks for your comment. In the revised manuscript, we analyzed the modules obtained from WGCNA to investigate gene modules that were highly correlated with trends in carotenoid metabolites during tomato fruit ripening. Two modules were enriched with genes involved in carotenoid and ethylene biosynthesis and signaling pathways by KEGG. We performed co-expressed gene network analysis using cytoscape, to identify transcription factors that may be involved in the carotenoids and ethylene biosynthesis and signaling pathways. The results of the analyses are described in the Results section of the revised manuscript and are presented in Supplementary Figure 3 and Supplementary Figure 4, (Line 184 - 186, 189 - 198).
- 3.Generally, you don't have a transition from color development to fruit ripening; you need to create additional graphs to support this transition. The current data curation gives the impression that these are two separate studies (color development via carotenoid accumulation and fruit ripening during ethylene signaling).
Response: Thanks for your comment. We combined metabolome and transcriptome data to perform WGCNA, with the objective of identifying the co-expressed genes and transcription factors related to carotenoid metabolism at different fruit developmental stages between the two varieties. In addition, KEGG analysis revealed, two pathways that exhibited a high degree of correlation with the observed trends in carotenoid metabolism between the two varieties at different developmental stages. The carotenoid biosynthesis pathway and the ethylene biosynthesis and signal transduction pathway, along with the genes associated with these pathways, exhibited a high degree of correlation in the observed variation in lycopene content. Furthermore, these genes were found to be co-expressed with certain transcription factors. This is why we analyzed the role of the ethylene biosynthesis pathway in regulating of fruit ripening and fruit color changes. These points are included in the Introduction and Results sections of the revised manuscript, respectively (Line 184 - 186, 189 - 198).

Reviewer 4 Report
Comments and Suggestions for Authors
The manuscript sounds interesting and may gather relevant contributions for fruit ripening and carotenoids metabolism. However, in the current way it is insufficient for publication. I highlighted several inconsistencies and concerns for clarification.
INTRODUCTION
The characterization of the EIN family is confusing because authors firstly mention that this family is small, but then describe it as a large family. This should be clarified (see lines 69 – 71).
MATERIAL AND METHODS
The statement “Six weeks old seedlings of both varieties were transplanted to the field and kept under natural growing conditions with sufficient light” is subjective. How much “sufficient light” means?
Carotenoid and ethylene metabolisms are major topics in the present study. Color determination and ethylene quantification are clearly a necessity; however, authors did not carry out these analyses. Why?
Which part of the fruit was sampled? This must be specified.
Carotenoid identification and quantification description are superficial. Authors should specify the conditions of injection and running used with LC/MS equipment (such as volume injected, time or running, mobile phase composition, flux, temperature, and so on). It is important to make the work reproducible.
In addition, which supply company and carotenoid standards were used to build the calibration curves?
Also, mRNA sequencing methodology is poorly described. Authors should review how libraries were constructed, the mode of sequencing (paired-end?), and read length.
Authors claim that HISAT2 provided the quantification of reads at genes and transcripts levels as normalized FPKM values. Then they mention that raw count data were subjected to DEGs analysis using DESeq2. I have an issue with this connection between HISAT2/DESeq2 given that HISAT2 provide normalized FPKM values, while DESeq2 input data must be raw counts (unnormalized), since DESeq2 workflow normalize that data before the inference of DEGs. Thus, I strongly recommend the authors to obtain raw read count using HTseq2, for example, and then analyze the DEGs again. In the current way, DEGs results and related analyses are unreliable and uncertain for me.
Besides, authors should indicate the comparisons established among developmental stages, whether DEGs inference were performed.
How PCA analysis was performed? I mean, which data and tool was used? There was any gene count filtering?
I could not find the primer sequences for SlLCYB2 and SlERF1 in the Table S1.
Which data were analyzed using GraphPad Prism 8.0? Authors should clarify it. In the way it is, It seems that every statistical analysis was carried out using this program.
RESULTS
The authors presented the heat map related to metabolite profile. I recommend them to present the content of each metabolite during fruit development and ripening stages and the statistical significance between varieties and among stages. This could be presented as a supplementary table.
Authors claim that “All clean reads were mapped to the SGN (Sol Genomics Network, Sol: 4.0) reference genome” (page 118-119). I would like to see the percentage of mapped reads for each replicate. Authors could present this information in supplementary figure 1.
DEGs list between varieties at each developmental stage should be included as supplementary material.
Figure 2C - Authors should indicate which stages and varieties were subjected to KEGG. It is unclear. I would like to see this enrichment for 40, 50 and 60 DPA between varieties.
Authors did not indicate which grid is representative of each variety in figures 4 and 5. This should be clarified.
Regarding RT-qPCR validation (Figure 4 and 5), I strongly recommend to authors to represent the relative expression data in log2FC, between the same comparisons established for RNA-seq DEGs inference. Thus, authors could exhibit RNA-seq and RT-qPCR side by side and also perform Pearson correlation analysis. No doubt, it would evidence RNA-seq robustness and reliability.
In the sentence “These genes are members of gene families, including, L-Methionine (Met), S-Adenosyl-L-methionine (SAM), 1-Aminocyclopropane-1-carboxylate (ACC)… (lines 208-210)” authors referred to gene family names, but included the name of substrate.
DEG mentioned in results and discussion sections (also presented in figures) must be sufficiently identified. I noticed that authors used only the gene family name and sometimes indicated specific gene members. However, the gene ID provided by genomic annotation must be shown in the figures. This identification is necessary so that readers may easily identify and perform further study based on the author's results.
Additionally, ACS, ACO and ERFs genes members up-regulated must be specified. In the current way, authors presented these results superficially. It is unclear which genes they are referring to. I mean, these genes must be named (ACS1, ACS2, ACS3, and so on) and their respective gene ID. Also, figure 5A is only cited at the end of these results.
Data availability statement must present the accession number where RNA-seq data were made publicly available.
DISCUSSION
In the sentence “The accumulation of carotenoids is regulated by the expression of genes encoding catalase enzymes in their pathways” (lines 259 – 260). I could not find a clear association between catalase and carotenoid biosynthesis. Could authors support this?
Authors mention that “Furthermore, transcriptomic and metabolomic data were utilized collectively to construct a co-expression network for WGCNA analysis (lines 276-277)”. It is unclear the type of data subjected to WGCNA analysis. The correlation was performed among genes or between metabolomic and transcriptomic data?
“Our transcriptome analysis revealed that the major transcriptional differences occurred at the breaker stages (40-50 DPA, lines 271-272)”. Authors should be more specific. Would it be from the breaker to light red/orange fruits?
Author Response
Manuscript ID: ijms-3022080
Type: Research Article
Title: Integrative analysis of metabolome and transcriptome reveals the mechanism of fruit ripening in tomato (Solanum lycopersicum)
Dear Editor:
Thank you for your cooperation. We have carefully reviewed the comments carefully and have made corrections that we hope will meet with approval.
All the corrections are marked in red in the revised manuscript.
Response to Reviewer #4
INTRODUCTION:
- The characterizationof the EINfamily is confusing because authors firstly mention that this family is small, but then describe it as a large family. This should be clarified (see lines 69 – 71).
Response: Thanks for your comments. In the Introduction section of the manuscript, we described the two ethylene response gene families, of which “EIN is a small family of transcription factors” and in the next sentence we described that “In contrast to EIN, ERF is a larger family of plant transcription factors”. The statement “larger family” is only used for the ERF family compared to EIN (Line 71 - 73).
MATERIALS AND METHODS:
- Thestatement “Six weeks old seedlings of both varieties were transplanted to the field and kept under natural growing conditions with sufficient light” is subjective.How much “sufficient light” means?
Response: Thanks for your comments. As we mentioned that “six week old seedlings of both varieties were transplanted to the field and kept under natural growing conditions”, here “sufficient light” means “natural sunlight”, (Line 437).
- Carotenoid and ethylene metabolisms are major topics in the present study. Color determination and ethylene quantification are clearly a necessity; however, authors did not carry out these analyses. Why?
Response: We thank the reviewers for their suggestion. In the preliminary study, we mainly focused on the trend of fruit carotenoid content by observing the phenotypic differences in fruit color between the two tomato varieties at different stages. We agree with the reviewer that color determination and ethylene quantification are key indicators of the ripening process, and their analysis will further improve our research results. However, fruit color determination and ethylene quantification require fresh sample material and due to the climatic limitation in Xinjiang regions, our research material is currently at the one-month-old seedling stage, and we will continue to perform this part in the subsequent in-depth study.
- Which part ofthe fruitwas sampled? This must be specified.
Response: Many thanks for your comment. In this study, the samples were taken from whole tomato berries, including the exocarap (skin) and flesh. The description has been added to the Materials & Method section of the main body of the paper (Line 442).
- Carotenoid identification and quantification description are superficial. Authors should specify the conditions of injection and running used with LC/MS equipment (such as volume injected, time or running, mobile phase composition, flux, temperature,and soon). It is important to make the work reproducible.
Response: Thanks to the reviewer's suggestion. To make this work reproducible, the details of the conditions of LC/MS equipment have been added to the Materils Methods section of the revised manuscript. Carotenoid content was determined using UPLC (ExionLC AD) and tandem mass spectrometery (MS/MS) (QTRAP 6500+, N). Chromatographic separations were performed on a YMC C30 column (3 µm, 2 mm × 100 mm) at 28°C. The flow rate was set at 0.8 mL/min and the mobile phase consisted of two different solvents, acetonitrile: methanol (1:3, v:v), 0.01% butylated hydroxytoluene (BHT) and 0.1% formic acid solution (solvent A) and methyl tert-butyl ether and 0.01% BHT solution (solvent B). The gradient programs of the solutions (solvent A: solvent B) were as follows; 100:0 V/V at 0 min, 100:0 v:v at 3 min, 30:70 v:v at 5 min, 5:95 v:v at 9 min, 100:0 v:v at 10 min, 100:0 v:v at 11 min, and total of 2 μL sample was used for the quantification process”, (Line 455 - 464).
- In addition, which supply company and carotenoid standards were used to build the calibration curves?
Response: Thanks for your comment. The carotenoids were quantified using calibration curves of 12 standards (0.001 μg/mL, 0.005 μg/mL, 0.01 μg/mLm, 0.05 μg/mL、0.1 μg/mL、0.5 μg/mL、1 μg/mL, 5 μg/mL, 10 μg/mL, 50 μg/mL, 100 μg/mL, 250 μg/mL, 400 μg/mL) and the standard curve equation and correlation coefficients for the detected substances have been added to the Supplementary Materials Supplementary Table 1, (Line 465 - 467).
- Also, mRNA sequencing methodology is poorly described. Authors shouldreviewhow libraries were constructed, the mode of sequencing (paired-end?), and read length.
Response: The mentioned methodology has been reviewed and a detailed description of the library construction, sequencing mode and read length has been included in the Materials and Method section of the revised manuscript (Line 470 - 477).
- Authors claim that HISAT2 provided the quantification of reads at genes and transcripts levels as normalized FPKM values. Then they mention that raw count data were subjected to DEGs analysis using DESeq2. I have an issue with this connection between HISAT2/DESeq2 given that HISAT2 provide normalized FPKM values, while DESeq2 input data must be raw counts (unnormalized), since DESeq2 workflow normalize that data before the inference of DEGs. Thus, I strongly recommend the authors to obtain raw read count using HTseq2, for example, and then analyze the DEGs again. In the current way, DEGs results and related analyses are unreliable and uncertain for me.
Response: Thanks for your valuable suggestion. In this study, HTSeq v0.6.125 was used to count the numbers of read mapped to each gene of the transcriptome data, and the description has been added in the Material and Methods section of the revised manuscript (Line480 - 481).
- Besides, authors should indicate the comparisons established among developmental stages, whether DEGs inference were performed.
Response: Thanks for the comment. To better understand the transcript levels of two tomato varieties during different fruit development and ripening stages, we counted and comapared the number of DEGs in tomato fruits at adjacent stages of the same variety as well as between the two different varieties at the same stage. The results of the analysis have been supplemented in the Results section of the revised manuscript with Figure S1 “Statistics of differentially expressed genes”, (Line 150 - 155).
- How PCA analysis was performed? I mean, which data and tool was used? There was any gene count filtering?
Response: Data from 36 samples (three replicates per group) from fruits of two tomato varieties at six periods were used for PCA analysis. The results of PCA were analyzed and visualized using R Studio software (https://www.rstudio.com/) with FactoMineR (We don’t have any gene count filtering in PCA analysis). The above has been added to the Materials and Methods section of the revised manuscript (Line 487 - 489).
- I could not find the primer sequences for SlLCYB2and SlERF1in the Table S1.
Response: We apologize for the missing sequences. The primer sequences for SlLCYB2 and SlERF1 have been added to Supplementary Table 7 in the revised manuscript.
- Which data were analyzed using GraphPad Prism 8.0? Authors should clarify it. In the way it is, It seems that every statistical analysis was carried out using this program.
Response: Thanks for your suggestion. GraphPad Prism 8.0 was used to statistically analyze the qRT-PCR results for significant differences, and this has been clarified in the revised manuscript (Line 512).
RESULTS:
- The authors presented the heat map related to metabolite profile. I recommend them to present the content of each metabolite during fruit development and ripening stages and the statistical significance between varieties and among stages. This could be presented as a supplementary table.
Response: Thanks to the reviewer's suggestion. In the revised manuscript, we have listed the carotenoids that differed in content between the two varieties during fruit development and ripening stages and the results are shown in Supplementary Table 2, (Line 111 - 113). In addition, the statistical significance of the metabolites between the two varieties and among the stages has been shown in Supplementary Table 3, (Line 118 - 125).
- Authors claim that “All clean reads were mapped to the SGN (Sol Genomics Network, Sol: 4.0) reference genome” (page 118-119). I would like to see the percentage of mapped reads for each replicate. Authors could present this information in supplementary figure 1.
Response: In the Materials and Methods section of the revised manuscript, the mentioned information on the percentage of mapped reads for each sample has been added Supplementary Table 4, (Line 137 - 138 ).
- DEGs list between varieties at each developmental stage should be included as supplementary material.
Response: As suggested by the reviewer, the list of all DEGs between varieties at each developmental stage has been added Supplementary Table 6 in the revised manuscript and described in the Results section (Line 147 - 150).
- Figure 2C - Authors should indicate which stages and varieties were subjected to KEGG. It is unclear. I would like to see this enrichment for 40, 50 and 60 DPA between varieties.
Response: We performed a KEGG enrichment analysis on all overlapping DEGs in two tomato varieties (i.e., WP190 & ZH108) at ripening stages (40, 50 and 60 DPA). (Including W40 VS Z40, W50 VS Z50, W60 VS Z60) (Line158 - 160, 166 - 167).
- Authors did not indicate which grid is representative of each variety in figures 4 and 5. This should be clarified.
Response: Thank you for the reviewer's point. In the revised manuscript, we have added some tabs in Figures 4 and 5 about which grid is representative of each sample.
- Regarding RT-qPCR validation (Figure 4 and 5), I strongly recommend to authors to represent the relative expression data in log2FC, between the same comparisons established for RNA-seq DEGs inference. Thus, authors could exhibit RNA-seq and RT-qPCR side by side and also perform Pearson correlation analysis. No doubt, it would evidence RNA-seq robustness and reliability.
Response: Thanks to the reviewer's suggestion. To verify the reliability of RNA-seq, we plotted the relative expression data in log2FC, and standardized it uniformly with RNA-seq data. We then performed Pearson's correlation analysis, and described the correlation results in the Results section of the revised manuscript Supplementary Figure 5, (Line 227 - 232, 282 - 284).
- In the sentence “These genes are members of gene families, including, L-Methionine (Met), S-Adenosyl-L-methionine (SAM), 1-Aminocyclopropane-1-carboxylate (ACC).(lines 208-210)” authors referred to gene family names, but included the name of substrate.
Response: In the Results section of the revised manuscript, we have added the specific names of the genes and the genes ID are listed in Supplementary Table 8, (Line 251 - 254).
- 8. DEG mentioned in results and discussion sections (also presented in figures) must be sufficiently identified. I noticed that authors used only the gene family name and sometimes indicated specific gene members. However, the gene ID provided by genomic annotation must be shown in the figures. This identification is necessary so that readers may easily identify and perform further study based on the author's results. Additionally, ACS, ACO and ERFs genes members up-regulated must be specified. In the current way, authors presented these results superficially. It is unclear which genes they are referring to. I mean, these genes must be named (ACS1, ACS2, ACS3, and so on) and their respective gene ID. Also, figure 5A is only cited at the end of these results.
Response:Thanks to the reviewers' suggestions, in the results section of the revised manuscript, we have described in detail the specific names of the up- and down-regulated genes and listed the IDs of all the DEGs involved in Figure 4 and Figure 5 in Supplementary Table 8, (Line 257, 259, 261, 268, 270).
- Data availability statement must present the accession number where RNA-seq data were made publicly available.
Response: The transcriptome raw data have been uploaded to NCBI (https://www.ncbi.nlm.nih.gov/) and the Project number has been mentioned in the Data availability statement of the revised manuscript (Line 561 - 562).
DISCUSSION:
- In the sentence “The accumulation of carotenoids is regulated by the expression of genes encoding catalase enzymes in their pathways” (lines 259 – 260). I could not find a clear association between catalase and carotenoid biosynthesis. Could authors support this?
Response: Thanks to the reviewer for the question, the catalytic enzymes mentioned in the discussion are not accurate enough, we would like to express that some of the genes encoding key enzymes in the carotenoid synthesis pathway, including phytoene synthase enzyme (PSY), phytoene desaturase (PDS), ζ-carotene desaturase (ZDS), prolycopene isomerase (CRTISO), ζ-carotene isomerase (ZISO), lycopene andδ-cyclase (LCYE) or lycopene β-cyclase (LCYB). Up- and down-regulated expression of these genes can regulate carotenoid levels by catalyzing carotenoid synthesis and catabolism. This point have been added in the Discussion section of the revised manuscript (Line 307 - 311).
- Authors mention that “Furthermore, transcriptomic and metabolomic data were utilized collectively to construct a co-expression network for WGCNA analysis (lines 276-277)”. It is unclear the type of data subjected to WGCNA analysis. The correlation was performed among genes or between metabolomic and transcriptomic data?
Response: In the Methods section we mentioned that “To identify genes associated with carotenoid synthesis, differentially expressed genes (DEGs) and differentially accumulated carotenoids (DACs) were detected in the six developmental stages of fruit for integrative analysis. Transcriptomic and metabolomic data were used together to construct a co-expression network for WGCNA analysis to analyze the correlation between transcript levels and carotenoid metabolic levels. This point has been added in the Discussion section of the revised manuscript (Line 313 - 318).
- “Our transcriptome analysis revealed that the major transcriptional differences occurred at the breaker stages (40-50 DPA, lines 271-272)”. Authors should be more specific. Would it be from the breaker to light red/orange fruits?
Response: Thanks to the reviewer's reminder. In the discussion section of the revised manuscript, we have added a detailed description of the breaker stages, in which the fruits of the two varieties were transformed to light red (WP190) and light orange (ZH108) colors (Line 325 - 328).

Round 2
Reviewer 1 Report
Comments and Suggestions for Authors
The authors diligently carried out revisions according to the reviewer's comments.
I recommend this manuscript for publication in its current form.
Congratulations on the excellent work!
Reviewer 2 Report
Comments and Suggestions for Authors
My comments were satisfactorily answered by authors. The manuscript is adequate to be accepted for publication in the present version,
Reviewer 3 Report
Comments and Suggestions for Authors
All my concerns have been successfully addressed. Hence, I recommend this manuscript for publication in present form.
Reviewer 4 Report
Comments and Suggestions for Authors
The new manuscript version was not sufficiently improved. There are concerns that still remain unsolved and authors responses just reinforced them.
Authors claim that “All clean reads were mapped to the SGN (Sol Genomics Network, Sol: 4.0) reference genome”. I did not see the percentage of mapped reads for each replicate in Table S4 as indicated by authors. I found interesting that all clean reads from both varieties have mapped to the reference genome. In my practical experience, I have never got or seen that achievement.
The major concern is related to gene expression analysis (DEGs). Authors confirmed in this new version that they used HISAT2 program for mapping, HTSeq v0.6.125 count the read numbers mapped to each gene and FPKM was used for the quantification of genes and transcripts at the gene/transcript level. The raw count data was subjected to differential expression analysis using the DESeq2 software. There are strong evidences that authors used normalized counts for DEGs analysis with DESeq2. Thus, DEGs results and related analyses are still unreliable and uncertain for me.